# Molecular Evidence of Raccoon Dog *(Nyctereutes procyonoides*) as a Natural Definitive Host for Several *Sarcocystis* Species

**DOI:** 10.3390/pathogens14030288

**Published:** 2025-03-15

**Authors:** Petras Prakas, Tamara Kalashnikova, Naglis Gudiškis, Donatas Šneideris, Evelina Juozaitytė-Ngugu, Dalius Butkauskas

**Affiliations:** Nature Research Centre, Akademijos Str. 2, 08412 Vilnius, Lithuania; tamara.kalashnikova@gamtc.lt (T.K.); naglis.gudiskis@gamtc.lt (N.G.); donatas.sneideris@gamtc.lt (D.Š.); evelina.ngugu@gamtc.lt (E.J.-N.); dalius.butkauskas@gamtc.lt (D.B.)

**Keywords:** *Sarcocystis*, raccoon dog, molecular identification, *cox1*, definitive host, epidemiology

## Abstract

*Sarcocystis* parasites infect a wide range of animals, including reptiles, birds, and mammals, and have complex two-host prey–predator life cycle. Sarcocysts are mainly found in the muscles of intermediate hosts, and oocysts sporulate in the intestines of the definitive host. The raccoon dog (*Nyctereutes procyonoides*), native to Asia and invasive in Europe, is a known disease carrier. However, studies on raccoon dogs in the transmission of *Sarcocystis* are scarce. Between 2019 and 2024, a total of 26 raccoon dog carcasses were collected in Lithuania. The results of a light microscopy examination indicated that 50% of the samples were positive for *Sarcocystis* spp. sporocysts and sporulated oocysts. Based on nested PCR and sequencing of *cox1*, 88.5% of the samples were positive for these parasites. Molecular analysis revealed the presence of 11 different *Sarcocystis* species. Eight species, including *S. alces*, *S. capracanis*, *S. hjorti*, *S. iberica*, *S. linearis*, *S. morae*, *S. tenella*, and *S. venatoria* were reported for the first time in raccoon dogs as definitive hosts. The identified *Sarcocystis* species were linked to intermediate hosts, such as cervids, wild boars, pigs, goats, and sheep. These findings suggest that raccoon dogs play a key role in the spread of *Sarcocystis*, particularly species infecting cervids.

## 1. Introduction

Protozoan parasites of the genus *Sarcocystis* Lankaster (1982) have a compulsory two-host life cycle that relies on a prey–predator relationship. These parasites are abundant, distributed worldwide, and infect a wide range of reptiles, birds, and mammals. The genus *Sarcocystis* is characterized by the formation of sarcocysts, mainly in the muscles of intermediate hosts (IHs). Sexual multiplication of parasites takes place in the intestines of the definitive hosts, and the sporocysts produced after the sporulation of the oocysts are released into the environment in feces. The IH becomes infected by consuming food or water contaminated with sporocysts, while the definitive host becomes infected by ingesting tissues containing mature sporocysts [1].

The raccoon dog, *Nyctereutes procyonoides* (Gray, 1834), is a widespread invasive canid species in northern, eastern, and central Europe [2]. It is commonly believed that the animal was introduced to the territory of Lithuania in 1948 and by 1960 had already invaded it [3]. Raccoon dogs were brought to Europe because of their valuable fur or as pets and either escaped or were intentionally introduced into the wild [4]. The raccoon dog is known as an ecological generalist, meaning that it is highly adaptable to a wide range of environmental conditions and has a variety of food preferences. They are omnivores, so their eating habits vary with the seasons and the environment in which they live [5,6]. Typical foods may include invertebrates, small mammals, amphibians, birds, carrion, insects, and plants [7]. The current population of raccoon dogs in Lithuania is estimated to be at least 10,000 individuals, which was counted by the number of road kills and hunted animals [8].

The raccoon dog is listed among the 100 worst alien species in Europe [9]. This invasive species can damage biological diversity and impact autochthonous ecosystems [10]. Raccoon dogs are omnivorous and can prey on native fauna, as well as compete with native predators for natural resources [2,10]. This invasive canid species is also characterized by its high behavioral plasticity, reproductive capacity, and adaptability to a wide range of environmental conditions, which has allowed it to successfully colonize Europe [2]. In addition, according to the enemy release hypothesis, an invasive host may have a competitive advantage over native species by leaving behind its natural enemies, including parasites [11]. The raccoon dog encounters and accumulates parasites found in newly colonized areas, and parasitological investigations are essential to identify these organisms. Furthermore, it must be taken into account that this invasive species can be a vector and potential reservoir for several illnesses [2].

Raccoon dogs are known to spread many zoonotic and pathogenic diseases [2,10,12,13,14,15] that can be harmful to both human and animal health. These invasive mammals can also serve as reservoirs for a variety of vector-borne diseases, including *Borreliella* Adeolu and Gupta (2015) spp. and *Rickettsia* da Rocha-Lima (1916) spp., and for a number of infectious agents, including rabies lyssavirus, canine distemper virus, SARS-CoV-2, *Trichinella* Railliet (1895) spp., *Baylisascaris procyonis* Stefanski and Zarnowski, 1951, and *Echinococcus multilocularis* Leuckart, 1863 [12,13,14,15,16]. *Trichinella* spp. have previously been detected in muscle tissues of raccoon dogs in Lithuania, Latvia, and Estonia [12,17]. In addition, cestodes (*Echinococcus multilocularis* Leuckart, 1863, *Mesocestoides* Vaillant (1863) spp., and *Taenia* Linnaeus (1758) spp.), nematodes (*Toxocara canis* Werner, 1782, *Capillaria* Zeder (1800) spp., and *Uncinaria stenocephala* Railliet, 1884), and trematodes (*Alaria alata* (Goeze, 1782) Krause, 1914 and *Opistorchis felineus* (Rivolta, 1884) Blanchard, 1895) were detected in Lithuanian raccoon dogs through detailed helminthological examination [17]. Some helminth species can cause serious illness in humans; for example, infection with *Trichinella* spp. can lead to death [18]. On the other hand, *Sarcocystis* spp. infection in carnivores as definitive hosts of these parasites are mainly asymptomatic. However, it has been shown that among *Sarcocystis* spp. forming sarcocysts in meat-producing livestock, species transmittable via canids are more pathogenic compared to those transmittable via other definitive hosts [1]. In this context, it is relevant to examine different canid species for their role in the transmission of various *Sarcocystis* species. Up to date among canids, dogs [19], gray wolves, *Canis lupus* (Linnaeus, 1758) [20,21], red foxes, *Vulpes vulpes* (Linnaeus, 1758) [22,23], arctic foxes, *Vulpes lagopus* (Linnaeus, 1758) [24], and golden jackals, *Canis aureus* (Linnaeus, 1758) [25] were mostly investigated as potential definitive hosts of *Sarcocystis* species. However, there is still a limited number of studies on raccoon dogs as definitive hosts of *Sarcocystis* spp.

The objective of our study was to evaluate the prevalence of *Sarcocystis* spp. in the small intestines of raccoon dogs collected in Lithuania by means of light microscopy and molecular methods and to identify parasite species using DNA sequence analysis.

## 2. Materials and Methods

### 2.1. Sample Collection and Isolation of Sarcocystis spp.

The research of the current study was conducted under the approval guidelines of the Ethics Committee of the Nature Research Centre (no. GGT-1). The Minister of Environment in Lithuania approves the rules on hunting (20 June 2002 no. IX-966) of the Republic of Lithuania, a list of game species and the time limits for hunting these animals. Based on this document, the hunting of the common raccoon dog is allowed all year round in Lithuania. A total of 26 raccoon dog carcasses were collected in Lithuania between 2019 and 2024. The collection sites of raccoon dogs examined are shown in Figure 1.

Different forms of *Sarcocystis* spp. from the intestinal samples of raccoon dogs were isolated using a previously described protocol [26]. The necessary modifications made during this procedure were already detailed in the earlier work by Prakas et al. [27].

### 2.2. Molecular Analysis

DNA extraction from the intestinal scrapings of raccoon dogs was conducted with the GeneJET Genomic DNA Purification Kit (Thermo Fisher Scientific Baltics, Vilnius, Lithuania) following the manufacturer’s recommendations.

Nested PCR (nPCR) was employed to amplify partial *cox1* sequences from the extracted DNA samples. The primers used in this study are listed in Table 1. A total of 16 *Sarcocystis* species were selected, including five that use farm animals as intermediate hosts: *Sarcocystis arieticanis* Heydorn, 1985; *Sarcocystis bertrami* Doflein, 1901; *Sarcocystis capracanis* Fischer, 1979; *Sarcocystis cruzi* (Hasselmann, 1926) Wenyon, 1926; and *Sarcocystis tenella* (Railliet, 1886) Moulé, 1886 [28,29,30,31]. Additionally, ten species that infect members of the Cervidae family were chosen: *Sarcocystis alces* Dahlgren and Gjerde, 2008; *Sarcocystis capreolicanis* Erber, Boch and Barth, 1978; *Sarcocystis gracilis* Ratz, 1906; *Sarcocystis hjorti* (Dahlgren and Gjerde, 2008) Dahlgren and Gjerde, 2010; *Sarcocystis linearis* Gjerde, Giacomelli, Bianchi, Bertoletti, Mondani and Gibelli 2017; *Sarcocystis iberica, Sarcocystis morae*, *Sarcocystis venatoria* Gjerde, Luzon, Alunda and de la Fuente 2017; *Sarcocystis pilosa* Prakas, Butkauskas, Rudaitytė-Lukošienė, Kutkienė, Sruoga and Pūraitė, 2016; and *Sarcocystis taeniata* Gjerde, 2014 [32,33,34,35,36,37]. Lastly, *Sarcocystis miescheriana* (Kühn, 1865) Labbé, 1899, which infects both wild boars and domestic pigs, was included [38].

The first round of nPCR was carried out in a final volume of 25 μL, comprising 12.5 μL of DreamTaq PCR Master Mix (Thermo Fisher Scientific, Vilnius, Lithuania), 0.5 μM of both forward and reverse primers, and 4 μL of extracted DNA and nuclease-free water to make up the remaining volume. The cycling conditions began with an initial denaturation at 95 °C for 5 min, followed by 35 cycles of 35 s at 94 °C, 45 s at the species-specific annealing temperature (depending on the primer pair), 55 s at 72 °C, and finishing with 5 min at 72 °C. The second round of amplification was also conducted in the reaction volume of 25 μL, comprising 12.5 μL of DreamTaq PCR Master Mix, 0.5 μM of both forward and reverse primers, 2 μL of the amplified product from the first nPCR step, and nuclease-free water up to 25 μL. Positive controls, consisting of DNA extracted from *Sarcocystis* spp. sarcocysts in previous studies, and negative controls, consisting of nuclease-free water instead of the DNA template, were used for the assessment of both nPCR rounds.

The amplified PCR products were assessed using 1% agarose gel electrophoresis. Positive amplicons were purified using ExoI and FastAP (Thermo Fisher Scientific Baltics, Vilnius, Lithuania) according to the manufacturer’s recommendations. Following this, purified products were subjected to Sanger sequencing using the same forward and reverse primers as for the nPCR. The Big-Dye^®^ Terminator v3.1 Cycle Sequencing Kit and 3500 Genetic Analyzer (Applied Biosystems, Foster City, CA, USA) were used to perform sequencing reactions according to the manufacturer’s recommendations. Obtained sequences were checked manually for the absence of any double peaks or poly signals.

### 2.3. Sequence Data Analysis

To compute intraspecific and interspecific genetic similarity values, in the current study, the obtained *cox1* sequences were compared with those of closely related *Sarcocystis* spp. using the Nucleotide BLAST online tool ([42]; accessed on 9 January 2025). Phylogenetic analysis was performed using MEGA v. 11.0.13 software [43]. Multiple sequence alignments were created using the MUSCLE algorithm incorporated into MEGA. Sequences included in the sequence alignment differed in nucleotide substitutions but not in deletions/insertions. Phylogenetic relationships of *Sarcocystis* spp. were assessed using the maximum likelihood method. Based on MEGA’s Find Best DNA/Protein Models (ML) function, Kimura 2 parameter + G was selected as the best fitting to all alignments analyzed. The robustness of the phylogeny was tested using the bootstrap method with 1000 replicates.

### 2.4. Statistical Data Analysis

The Clopper–Pearson method was used to compute 95% confidence intervals (CIs) for the frequency of listed *Sarcocystis* spp. in definitive host animals collected for several years. A Chi-squared test was used to evaluate differences in the detection rates of *Sarcocystis* species in the examined raccoon dogs. Statistical tests were carried out using the Quantitative Parasitology 3.0 software [44].

## 3. Results

### 3.1. Microscopical Examination of Sarcocystis spp. Sporocysts/Oocysts from the Intestines of Raccoon Dogs

Under light microscopy, oocysts and/or sporocysts of *Sarcocystis* spp. were detected in 13 out of 26 intestinal samples (50.0%; 95% CI = 36.9–76.7) from raccoon dogs. In contrast, *Sarcocystis* spp. were identified in 23 out of 26 tested individuals (88.5%; 95% CI = 69.8–97.6) using molecular methods. The observed detection rates indicate a significantly higher efficiency of molecular methods compared to microscopy (χ^2^ = 9.03, *p* = 0.003).

Free sporocysts were observed in all samples where *Sarcocystis* parasites were detected microscopically, whereas in eight samples sporulated oocysts were seen. The number of *Sarcocystis* stages per 24 × 24 mm coverslip area ranged from 3 to 175 (73.1 ± 61.4). In most of the intestinal samples, free sporocysts were more frequently detected than sporulating oocysts. Sporocysts of *Sarcocystis* spp. measured 11.1–17.6 × 7.1–10.2 μm (14.0 ± 1.4 × 8.4 ± 0.9 μm; n = 222), whereas ellipsoidal sporulated oocysts were thin-walled and measured 14.6–23.2 × 12.0–18.1 μm (18.6 ± 3.0 × 14.8 ± 1.9; n = 31) (Figure 2). The identification of *Sarcocystis* species was conducted by further molecular analysis.

### 3.2. Molecular Identification of Sarcocystis Species

Using species-specific nested PCR and a subsequent sequencing approach, we have tested the presence of 16 *Sarcocystis* species in the intestinal samples of raccoon dogs. Overall, 11 different *Sarcocystis* species, i.e., *S*. *alces*, *S*. *capracanis*, *S*. *capreolicanis*, *S*. *gracilis*, *S*. *hjorti*, *S*. *iberica*, *S*. *linearis*, *S*. *miescheriana*, *S*. *morae*, *S*. *tenella,* and *S*. *venatoria* have been identified (Table 2). On the contrary, the designed primers failed to produce positive amplicons when applied to *S. arieticanis*, *S. bertrami*, *S. cruzi*, *S. taeniata*, and *S. pilosa*. By using V2ibeven1/V2ibeven2 primers, theoretically designed to amplify *S*. *iberica* and *S*. *venatoria cox1* fragments, *S*. *iberica* was detected in three samples, and another tested species was found in one sample. However, only one species, *S*. *linearis,* was identified with the help of V2taelin1/V2taelin2 primers *in silico* designed to amplify *cox1* of either *S*. *linearis* or *S*. *taeniata*. After the resulting sequences were truncated by discarding the primer-binding nucleotides, the lengths of the analyzed sequences varied between 207 and 617 bp (Table 2). One to five *cox1* haplotypes were identified in the *Sarcocystis* species detected. Our generated sequences showed very high similarity compared to other sequences of the same *Sarcocystis* species available in GenBank (96.2–100%). Despite the short DNA fragments compared, for each tested *Sarcocystis* species, the calculated values of intraspecific genetic similarity and interspecific genetic similarity did not overlap, indicating the correctness of the method used in this study.

For some species, the differences between the intraspecific and interspecific genetic variability values obtained were relatively small. Specifically, the resulting sequences of *S. capracanis*, *S. capreolicanis*, *S. hjorti*, *S. iberica*, *S. linearis*, *S. tenella,* and *S. venatoria* showed 91.5–97.7% similarity with those of most closely related species. Therefore, phylogenetic analyses were carried out for a conclusive identification of these species. In phylograms, our sequences clustered with those of the same species obtained from the GenBank, confirming the identity of the *Sarcocystis* species established (Figure 3). Based on the *cox1* fragments examined, *S*. *linearis* was a sister species to *S*. *taeniata* (Figure 3a), *S*. *capracanis* was a sister species to *S*. *tenella* (Figure 3b,e), *S*. *capreolicanis* showed the closest relationships with *S*. *alceslatrans*, *S*. *gjerdei,* and *S*. *pilosa* (Figure 3c), *S*. *hjorti* was most closely related to *S*. *pilosa* (Figure 3d), and *S*. *iberica* clustered with *S*. *venatoria* (Figure 3f). In general, the species identified were most closely related to *Sarcocystis* spp. using members of the Bovidae or Cervidae family as their IHs and canids as their definitive hosts.

### 3.3. Distribution of Sarcocystis Species in Raccoon Dog Samples Analyzed

By using molecular methods, we have identified the *Sarcocystis* species utilizing both domestic and wild animals as IHs in the intestinal samples of raccoon dogs. Among the molecularly confirmed *Sarcocystis* species, the most prevalent ones were *S. gracilis* (46.2%), *S. capreolicanis* (42.3%), and *S. hjorti* (38.5%), which employ Cervidae as IHs and *S. miescheriana* (38.5%), forming sarcocysts in pigs/wild boars (Table 3). The analysis revealed that *S. gracilis* was found to be significantly more prevalent in raccoon dogs than several other *Sarcocystis* species, including *S. alces* (χ^2^ = 4.28, *p* < 0.05), *S. iberica* (χ^2^ = 7.59, *p* < 0.0001), *S. tenella* (χ^2^ = 4.28, *p* < 0.05), and *S. venatoria* (χ^2^ = 12.41, *p* < 0.00001). Similarly, *S. capreolicanis* showed significantly higher detection rates compared to *S. iberica* (χ^2^ = 10.83, *p* < 0.00001) and *S. venatoria* (χ^2^ = 6.26, *p* < 0.05). In addition, *S. hjorti*, *S. miescheriana*, and *S. morae* were detected more frequently than *S. venatoria* and *S. iberica* (*p* < 0.05). On the other hand, *S. venatoria* was found to be less common than both *S. capracanis* (χ^2^ = 4.13, *p* < 0.05) and *S. linearis* (χ^2^ = 6.58, *p* < 0.05).

Among 23 *Sarcocystis* spp.-positive raccoon dogs, a single *Sarcocystis* species was identified only in one animal, while in the rest of the samples analyzed, two to seven different *Sarcocystis* species were established. In most cases, two to four different *Sarcocystis* species were present within a single animal specimen, accounting for 69.2% of the samples investigated (18/26). The most prevalent parasite species combinations involved distinct *Sarcocystis* species from the Cervidae family, as this was recorded in 34.6% of animals (9/26). In five samples (19.2%), we found at least one species of *Sarcocystis* using Caprinae, Cervidae, and Suidae as their IHs. For instance, *S*. *capracanis* and *S*. *tenella* (IH: Caprinae), *S*. *hjorti* (IH: Cervidae), and *S*. *miescheriana* (IH: Suidae) were identified in one of the samples.

## 4. Discussion

Raccoon dogs have been recognized as one of the most successful invasive carnivorous species after their introduction to the European continent from East Asia [45]. Their diverse diet and adaptability to various environments allowed them to spread not only across Lithuania but throughout eastern, central, and northern parts of Europe, competing with native species [4,46]. Despite this, data on the role of raccoon dogs in spreading *Sarcocystis* parasites are scarce, as is information on the diversity of these parasites within this host, particularly in their native range [47,48]. Traditionally, identification of *Sarcocystis* parasites in definitive hosts relied on transmission experiments using experimental animals from the Canidae family, such as dogs or foxes, and from the Felidae family, such as cats [49,50,51]. Nowadays, due to ethical issues, the use of laboratory experiments on predatory mammals is limited. Therefore, molecular methods have become increasingly popular for identifying *Sarcocystis* species in fecal or mucosal scraping samples from wild predators infected under natural conditions [26,52,53].

During our investigation, molecular analysis revealed that 88.5% (23/26) of the intestinal samples from raccoon dogs were positive for *Sarcocystis* parasites, while microscopic analysis showed a lower detection rate of 50.0% (13/26). In many studies, molecular methods have been found to give higher detection rates of *Sarcocystis* spp. than microscopy. [23,54]. The scarcity and small size of excreted sporocysts in natural infections make the microscopic approach challenging for scientists without prior experience observing *Sarcocystis* spp. in the mucosal scrapings or feces of the definitive host [55]. Additionally, differences in detection rates may be attributed to the higher sensitivity of molecular methods, which can detect parasitic gDNA present in lower concentrations [53].

Based on the results of microscopical methods, the occurrence of *Sarcocystis* spp. in the intestines of raccoon dogs across Europe is generally similar, with 66.7% (2/3) found in the large intestines of raccoon dogs from the Czech Republic [23] and 52.6% (20/38) and 53.8% (7/13) in the small intestines of raccoon dogs from Germany [56] and Lithuania [22], respectively. Interestingly, lower detection rates of *Sarcocystis* parasites were recorded in the intestinal scrapings of foxes, also members of the Canidae family. Only 20.0% (4/20) of intestinal samples from red foxes in Lithuania [22], 38.0% (19/50) in Germany [56], 30.0% (6/20) in Switzerland [57], and 39.4% (13/33) of intestinal samples from Pampas foxes, *Lycalopex gymnocercus* (Fischer, 1814), in Argentina [58] tested positive for *Sarcocystis*. The relatively higher detection rates of *Sarcocystis* spp. in raccoon dogs’ intestinal scrapings compared to other Canidae species may be linked to their dietary preferences. However, all these species—raccoon dogs, red foxes, and others—consume similar diets, including small and medium to large mammals, ungulates, and carrion [59,60,61]. Therefore, further research involving various Canidae species within the same geographic area is needed to compare detection rates of *Sarcocystis* spp. among the members of the Canidae family.

Experimental studies confirmed raccoon dogs to be definitive hosts for at least four *Sarcocystis* species: *S. cruzi* (IH: cattle) [47], *Sarcocystis grueneri* Yakimoff and Sokoloff, 1934 (IH: Cervidae/reindeer (*Rangifer tarandus*)) [62], *S. miescheriana* (IH: wild boars and pigs) [47], and *Sarcocystis tarandivulpes* Gjerde, 1984 (IH: Cervidae/reindeer (*Rangifer tarandus*)) [62]. Through the application of molecular methods, raccoon dogs were found to be definitive hosts for four additional *Sarcocystis* species, *S. capreolicanis* (IH: Cervidae/roe deer (*Capreolus capreolus*)), *S. gracilis* (IH: Cervidae/roe deer (*Capreolus capreolus*)) [56], *Sarcocystis lutrae* Gjerde and Josefsen, 2015 (IH: carnivores of families Canidae, Mustelidae and Procyonidae) [63], and *Sarcocystis rileyi* (Stiles, 1893) Dubey, Cawthorn, Speer and Wobeser, 2003 (IH: birds of family Anatidae) [22]. Our study provides new evidence that raccoon dogs can serve as definitive host for eight *Sarcocystis* species, all of which were previously unidentified in this host. This includes six species (*S. alces*, *S. hjorti*, *S. iberica*, *S. linearis*, *S. morae*, and *S. venatoria*) that typically use Cervidae as IHs and two species (*S. capracanis* and *S. tenella*) that infect members of subfamily Caprinae.

Some *Sarcocystis* species are zoonotic and pose a significant health risk to humans. Currently, humans can act as definitive hosts for three species: *S. heydorni* and *S. hominis,* transmitted through cattle, and *S. suihominis*, which spreads through wild boars and pigs [64,65,66]. Furthermore, recent studies indicate that ingesting raw meat containing a high concentration of *Sarcocystis* spp. sarcocysts may result in toxic effects. In Japan, there have been documented instances where individuals developed temporary gastrointestinal issues, including nausea, abdominal discomfort, diarrhea, and vomiting, within a few hours of consuming contaminated horsemeat carrying *Sarcocystis fayeri* Dubey, Streitel, Stromberg and Toussant, 1997, sarcocysts or raw venison contaminated with *Sarcocystis truncata* (Gjerde, 1984) Gjerde, 2014 [67,68,69]. In addition to the human health impact, *Sarcocystis* species also affect various animal hosts, particularly ruminants. Occasionally, *S. tenella* and *S. arieticanis* infections in sheep can result in miscarriage or acute disease during the early stages, with chronic effects, such as reduced productivity and persistent illness, becoming evident in the later stages of infection [70,71]. In pigs, acute infections with *S. miescheriana* can lead to serious symptoms such as loss of appetite, fever, cyanosis, heart inflammation, liver damage, and kidney issues [72]. Similar pathological changes, such as fever, miscarriage, or even death, have been observed in goats with acute *S. capracanis* infections [73], though no zoonotic *Sarcocystis* species have been detected in small ruminants to date. Data on the pathogenicity of *Sarcocystis* species in cervids (Cervidae) are extremely limited. Despite that, a study in Switzerland identified *S. hjorti* as a cause of eosinophilic fasciitis in red deer (*Cervus elaphus*), highlighting the need for further research on its impact on wildlife health [57]. Evidence suggests that *Sarcocystis* spp. transmitted by canids or primates are generally more pathogenic than those spread by felids. In the present study, we have identified *S*. *capracanis*, *S*. *hjorti*, *S*. *miescheriana*, and *S*. *tenella,* which can be pathogenic to their respective IHs, indicating that raccoon dogs may contribute to the spread of pathogenic *Sarcocystis* species (Figure 4).

Raccoon dogs are considered medium-sized predators, and the winter season plays a crucial role in their survival [74,75]. Of all the species in the Canidae family, this is the only one that demonstrates winter lethargy in regions with severe winter conditions [6]. Although they typically hibernate, raccoon dogs in Lithuania remain active year-round during milder winters [60]. In this study, we identified multiple *Sarcocystis* species, with their IHs belonging to the Caprinae subfamily, Cervidae, and Suidae families, showing the importance of this predator in the transmission of various *Sarcocystis* spp. During previous studies, it was observed that active raccoon dogs often visit an increased number of artificial wild boar-feeding sites in Northern Europe [76], which may explain the relatively high detection rate of *S*. *miescheriana* in our study. However, the majority of *Sarcocystis* spp. detected during our study infect Cervidae as IHs. This preference may be linked to raccoon dogs’ avoidance of areas with high human activity, as they tend to favor natural habitats, particularly mixed forests in Lithuania [77]. Other members of the Canidae family prevalent in Lithuania, red foxes and gray wolves, are known to prey on livestock, particularly in rural habitats. These predators often target farm animals, including poultry, sheep, and cattle, when natural prey is scarce or when they are in close proximity to human settlements [78,79]. This feeding behavior is further supported by studies from Switzerland and Germany, which found that red foxes were more commonly infected with *Sarcocystis* spp. whose IHs are farm animals, while raccoon dogs were predominantly infected with species whose IHs are Cervidae [56,57].

## 5. Conclusions

The prevalence of *Sarcocystis* spp. in this study was notably higher when detected using molecular methods (88.5%) compared to microscopy (50.0%). The comprehensive analysis, incorporating microscopical, molecular, and phylogenetic techniques, revealed that raccoon dogs in Lithuania were infected with 11 distinct *Sarcocystis* species. Among these, *S. alces*, *S. capracanis*, *S. hjorti*, *S. iberica*, *S. linearis*, *S. morae*, *S. tenella*, and *S. venatoria* were identified in raccoon dogs as natural definitive hosts for the first time worldwide. Of the *Sarcocystis* species detected, some can be pathogenic to farm animals and cervids. Additionally, raccoon dogs more frequently harbored *Sarcocystis* species with wild animals as their IHs than those associated with farm animals.

## Figures and Tables

**Figure 1 pathogens-14-00288-f001:**
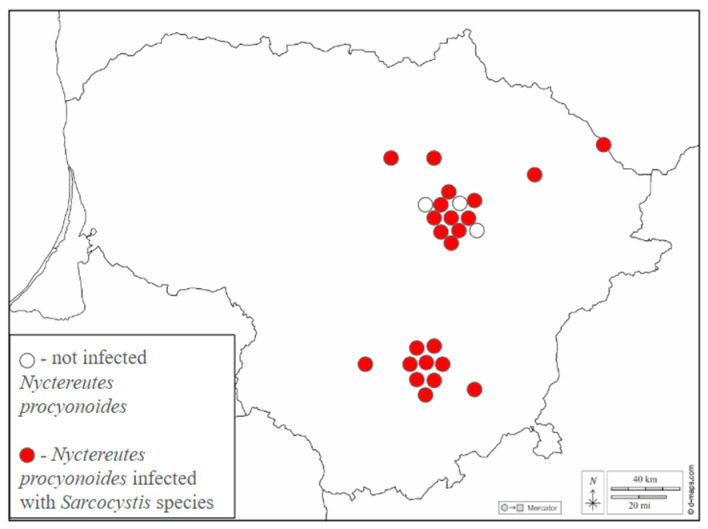
Detection of *Sarcocystis* spp. in the raccoon dogs from Lithuania. Circles indicate location where animals were hunted. Red circles represent *Sarcocystis* spp.-positive animals, while empty circles indicate individuals in which tested parasite species have not been established.

**Figure 2 pathogens-14-00288-f002:**
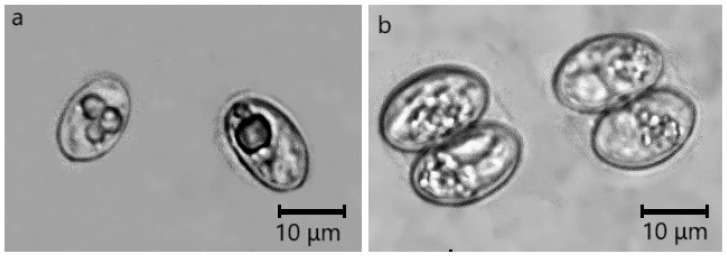
Sporocysts/sporulated oocysts found in the intestinal mucosa of raccoon dogs. (**a**) Sporocysts. (**b**) Sporulated oocysts.

**Figure 3 pathogens-14-00288-f003:**
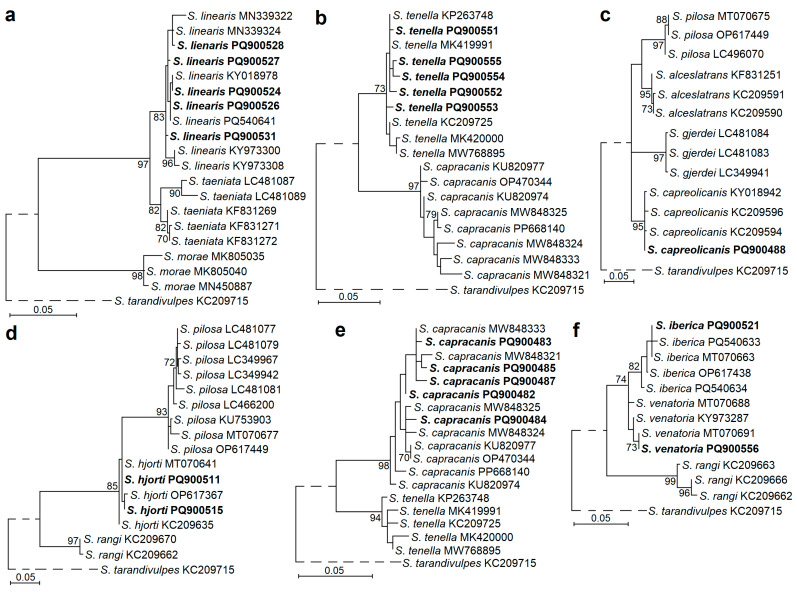
Maximum likelihood phylogenetic trees showing phylogenetic placement of *S*. *linearis* (**a**), *S*. *tenella* (**b**), *S*. *capreolicanis* (**c**), *S*. *hjorti* (**d**), *S*. *capracanis* (**e**), *S*. *iberica,* and *S*. *venatoria* (**f**). Phylograms were constructed using *cox1* sequences and scaled according to the branch length. *Sarcocystis tarandivulpes* was chosen as an outgroup for all analyses. Bootstrap values ≥ 70 are displayed next to branches. GenBank accession numbers of sequences are shown after the species name. Sequences generated in the present study are indicated in boldface.

**Figure 4 pathogens-14-00288-f004:**
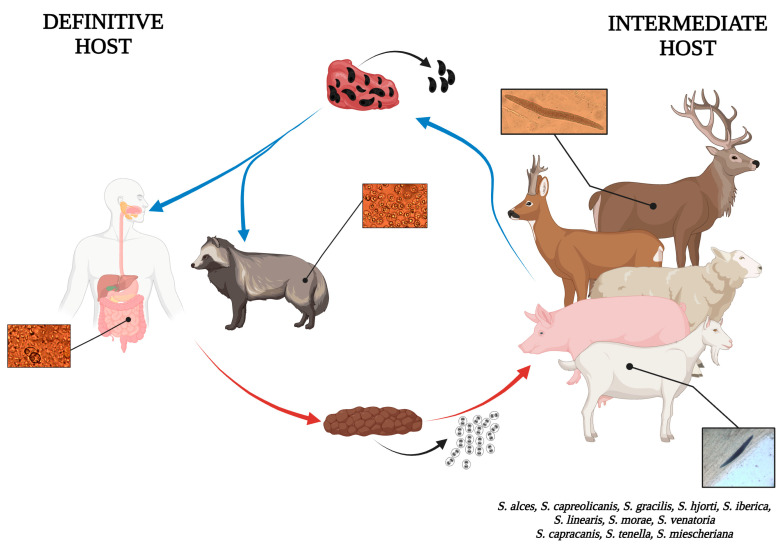
Schematic view of the involvement of raccoon dog in the life cycle of *Sarcocystis* spp. (Image created in BioRender. Gudiškis, N. (2025) https://BioRender.com/j04o051, accessed on 11 March 2025).

**Table 1 pathogens-14-00288-t001:** PCR primers used in this study.

Species	Primer Name	Sequence	Reference
1st step			
*Sarcocystis* spp.	SF1	ATGGCGTACAACAATCATAAAGAA	Dubey, 2013 [39]
	SsunR3	CCGTTGGWATGGCRATCAT	Marandykina-Prakienė et al., 2022 [40]
2nd step			
*S. alces*	V2alc3	CCTAGGTACCGTGCTCTTTGATG	Present study
	V2alc4	CTTCGAGGCCAGTAGTTACCATA	
*S. arieticanis*	Arie7F	TAATTTCCTCGGTACTGTACTGTTTG	Marandykina-Prakienė et al., 2023 [31]
	Arie7R	TACTTACGCATTGCGATATTACG	
*S. bertrami*	V2ber7	CCCCACTCAGTACGAACTCC	Baranauskaitė et al., 2022 [41]
	V2ber8	ACTGCGATATAACTCCAAAACCA	
*S*. *capracanis*	V2ca3	ATACCGATCTTTACGGGAGCAGTA	Marandykina-Prakienė et al., 2023 [31]
	V2ca4	GGTCACCGCAGAGAAGTACGAT	
*S*. *capreolicanis*	V2capreo1	CATCGTAGAGCCCCGTACTC	Present study
	V2capreo2	ACCGCTATACGCTGGAGCTG	
*S. cruzi*	V2cr7	CAATGTGCTGTTTACGCTCCA	
	V2cr8	TCGTACAGGCCCGTAGTTAG	
*S*. *gracilis*	V2gr9	GTGCTCGGGGCAGTGAAC	
	V2gr10	GCCAGTAGTCATCATGTGGTGT	
*S*. *hjorti*	V2hjo1	AAGGTACACGGCATTGTTCAC	
	V2hjo2	GAAAACTACCCTGCCGCCTA	
*S*. *iberica* *S*. *venatoria*	V2ibeven1	ATGGGCCATTATATTTACTGCTCTG	
	V2ibeven2	GCCGCCAAAAACTACCTTACC	
*S*. *miescheriana*	V2mie5	TCCTCGGTATTAGCAGCGTACTG	Baranauskaitė et al., 2022 [41]
	V2mie6	ATTGAAGGGCCACCAAACAC	
*S*. *morae*	V2mor1	GTGTGCTTGGATCGGTCAAC	Marandykina-Prakienė et al., 2023 [31]
	V2mor2	GCCGAATACCGGCTTACTTC	
*S. pilosa*	V2pil3	GTTAATTTCCTGGGCACAGTGTT	Present study
	V2pil4	CGAAAACTACTCTGCCGCCTAC	
*S*. *taeniata* *S*. *linearis*	V2taelin1	CGTAGACTGCATGACGACTTACAA	
	V2taelin2	CAAAGATGGATTTGCTGCCTA	
*S*. *tenella*	Ten8F	ATACCGCTCTACGCTGGATCTAC	Marandykina-Prakienė et al., 2023 [31]
	Ten8R	TACCGCTCTACGCTGGATCTAC	

**Table 2 pathogens-14-00288-t002:** The comparison of in the present study obtained *cox1* sequences with those of various *Sarcocystis* species available in GenBank.

*Sarcocystis* Species	GenBank Acc. No.	The Length of the Fragment	No. of Haplotypes	Similarity with the Same Species	Similarity with Most Closely Related Species
*S. alces*	PQ900477–PQ900481	352 bp	2	97.2–100%	*S*. *gracilis* 85.0–86.7 [QC = 96%]
*S. capracanis*	PQ900482–PQ900487	284 bp	5	97.2–100%	*S*. *tenella* 91.5–94.3%
*S. capreolicanis*	PQ900488–PQ900498	376 bp	1	99.5–100%	*S*. *alceslatrans* 95.0–95.5%
*S. gracilis*	PQ900499–PQ900510	371 bp	2	98.7–100%	*S*. *capracanis* 83.0–84.9%
*S. hjorti*	PQ900511–PQ900520	227 bp	2	96.7–100%	*S*. *pilosa* 92.5–94.3%
*S. iberica*	PQ900521–PQ900523	207 bp	1	99.5–100%	*S*. *venatoria* 95.7–97.1%
*S. linearis*	PQ900524–PQ900531	617 bp	5	98.5–100%	*S*. *taeniata* 96.8–97.7%
*S. mieshceriana*	PQ900532–PQ900541	315 bp	3	98.1–100%	*S*. *rangiferi* 75.6–76.9% [QC = 96%]
*S. morae*	PQ900542–PQ900550	292 bp	2	96.2–100%	*S*. *cervicanis* 83.6–85.3%
*S. tenella*	PQ900551–PQ900555	373 bp	5	98.7–100%	*S*. *capracanis* 91.7–93.8%
*S. venatoria*	PQ900556	207 bp	1	97.6–100%	*S*. *iberica* 95.7–96.1%

QC—query coverage.

**Table 3 pathogens-14-00288-t003:** Detection rates of identified *Sarcocystis* species in raccoon dogs.

*Sarcocystis* Species	Intermediate Host	*n*	Detection Rate, % (95% CI)
*S*. *alces*	Cervidae: moose (*Alces alces*)	5	19.2 (6.6–39.4)
*S*. *capreolicanis*	Cervidae: roe deer (*Capreolus capreolus*)	11	42.3 (23.4–63.1)
*S*. *gracilis*	Cervidae: roe deer	12	46.2 (26.6–66.6)
*S*. *hjorti*	Cervidae: moose, red deer (*Cervus elaphus*), sika deer (*Cervus nippon*)	10	38.5 (20.22–59.43)
*S*. *iberica*	Cervidae: red deer, sika deer	3	11.5 (2.4–30.2)
*S*. *linearis*	Cervidae: moose, red deer, roe deer, sika deer,	8	30.8 (14.3–51.8)
*S*. *morae*	Cervidae: fallow deer (*Dama dama*), red deer, sika deer,	9	34.6 (17.2–55.7)
*S. venatoria*	Cervidae: red deer	1	3.9 (0.1–19.6)
*S*. *miescheriana*	Suidae: pig and wild boar (*Sus scrofa*)	10	38.5 (20.2–59.4)
*S*. *capracanis*	Caprinae: goat (*Capra hircus*), Barbary sheep (*Ammotragus lervia*), European mouflon (*Ovis aries musimon*)	6	23.1 (09.0–43.7)
*S*. *tenella*	Caprinae: sheep (*Ovis aries*), argali (*Ovis ammon*), Barbary sheep, European mouflon, Tatra chamois (*Rupicapra rupicapra tatrica*)	5	19.2 (6.6–39.4)

*n*—number of infected animals.

## Data Availability

The *cox1* sequences of *Sarcocystis* spp. from intestinal mucosa scrapings of raccoon dogs were submitted in the GenBank database with accession numbers PQ900477–PQ900556.

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
