# Peer review of "Molecular Evidence of Raccoon Dog (Nyctereutes procyonoides) as a Natural Definitive Host for Several Sarcocystis Species"

_pathogens, 2025, doi:10.3390/pathogens14030288_

Round 1
Reviewer 1 Report
Comments and Suggestions for Authors
In this manuscript by Prakas, Kalashnikova and colleagues, the authors established a limited paper regarding the sarcocystis infesting in raccoon dogs. Personally, the manuscript provided only microscopic and phylogenetic results. The authors should include more significant findings before publication. Currently, I do not think that the context is proficient for publishing in the decent journal, pathogens.
Author Response
Point 1: In this manuscript by Prakas, Kalashnikova and colleagues, the authors established a limited paper regarding the Sarcocystis infesting in raccoon dogs. Personally, the manuscript provided only microscopic and phylogenetic results. The authors should include more significant findings before publication. Currently, I do not think that the context is proficient for publishing in the decent journal, Pathogens.
RESPONSE: Thank you for your remarks. In the Results section, we present not only microscopy and phylogenetic analysis findings but also molecular method results. Moreover, our study has established that the raccoon dog is a natural definitive host for as many as eight Sarcocystis species. We have added new information to the whole article, emphasising that our findings provide significant insights into the host-parasite relationship and contribute to the broader understanding of the diversity of Sarcocystis species. It should be noted that our study showed new definitive hots for even eight Sarcocystis species, i.e. it was for the first time worldwide confirmed that S. alces, S. capracanis, S. hjorti, S. iberica, S. linearis, S. morae, S. tenella, and S. venatoria might employ raccoon dogs as their definitive hosts.
Reviewer 2 Report
Comments and Suggestions for Authors
1) Overall, the present research work seems to be great.
2) However, the present authors could be molecular biologist, morphological or microscopical examinations results were poor in general. For example, the oocyst photos from the intestinal mucosa with the cell cultured methods will be absolutely valuable for (veterinary) medical education. Hence, they had better describe such information to both M&M and result parts.
3) They said that the raccoon dogs have been recognized as one of the most successful invasive species from East Asia. Hence, there could be many similar works in such aria including China, Russia and Japan. But the present authors did not refer to them. Probably, although there are language barrier among the literatures, they have to find that, and after them, add to their introduction and discussion parts.
Author Response
Point 1: However, the present authors could be molecular biologist, morphological or microscopical examinations results were poor in general. For example, the oocyst photos from the intestinal mucosa with the cell-cultured methods will be absolutely valuable for (veterinary) medical education. Hence, they had better describe such information to both M&M and result parts.
RESPONSE: Thank you for your remark. We agree that incorporating such methods, as suggested, to identify Sarcocystis forms in cell cultures would be beneficial for both veterinary and medical applications. However, since our study does not involve these techniques, we are unable to provide the requested material. Furthermore, we do not acquire equipment for suggested methods.
Point 2: They said that the raccoon dogs have been recognized as one of the most successful invasive species from East Asia. Hence, there could be many similar works in such aria including China, Russia and Japan. But the present authors did not refer to them. Probably, although there are language barrier among the literatures, they have to find that, and after them, add to their introduction and discussion parts.
RESPONSE: Thank you for your remark. To the best of our knowledge, although raccoon dogs are successful invasive species originating from East Asia, data on Sarcocystis spp. in these hosts remain scarce. We have added some new information to the discussion part (Line 238-240).
Reviewer 3 Report
Comments and Suggestions for Authors
The peer-reviewed manuscript deals with morphological and molecular study of Sarcocystis spp. from invasive species, raccoon dog inhabiting Lithuania and and correlate the obtained results with molecular data of Sarcocystis spp. from intermediate hosts - ungulates. This work provides new data on the role of raccoon dogs in the distribution of Sarcocystis spp. The manuscript is a well-thought-out and clearly structured article. The conclusions are confirmed by the results obtained by the authors. The article is beautifully illustrated. In general, the article leaves a favorable impression after reading it. The manuscript meets purposes and objectives of the Pathogens and can be published.
I have only minor remarks about this article:
- The title of the work is not clearly defined. The title should be concise, clearly reflecting the essence of the work. I suggest that the authors shorten the title of the manuscript a little: “The raccoon dog Nyctereutes procyonoides as a natural definitive host for several Sarcocystis species”
- In Table 1, it is necessary to add a separate column for the authors of sequences, where in addition to the reference number, the authors' surnames must be given. For primers obtained in this study, write "this study" in this column.
Again, the Table 2 lacks references to the authors of the sequences. This needs to be corrected.
Information about bioethics (Line 64-68) should be provided at the end of the article in the appropriate section. It is unnecessary here.
Line 12 –correct word - “hosts”. It is necessary to add information about the intermediate hosts of these parasites here.
The first table in the results is not needed. Part of the numerical material is already given in the text of the article. The remaining figures should also be given in the text. Then the numbering of the tables in the article will be corrected 😊
At the first mention of genus or species its full Latin name with the author and year of description should be given; in relation all animal species and Sarcocystis spp. (lines 26, 35, 55, 56, etc.). This should also be done in Tables 1 and 3. In Abstract Latin species name must be written in full.
Authors must submit the manuscript in accordance with the rules for formatting MDPI articles (Article title, all words in which must be capitalized).
The manuscript must be published in Pathogens after minor revision.
Author Response
Point 1: The title of the work is not clearly defined. The title should be concise, clearly reflecting the essence of the work. I suggest that the authors shorten the title of the manuscript a little: “The raccoon dog Nyctereutes procyonoides as a natural definitive host for several Sarcocystis species”
RESPONSE: Thank you for your remark, the title of the article has been changed accordingly.
Point 2: In Table 1, it is necessary to add a separate column for the authors of sequences, where in addition to the reference number, the authors' surnames must be given. For primers obtained in this study, write "this study" in this column.
RESPONSE: We have added a new column, titled “Reference” for the Table 1, indicating the first author surname for the referenced publication. Additionally, primers designed during our study are highlighted under “present study”. Thank you.
Point 3: Again, the Table 2 lacks references to the authors of the sequences. This needs to be corrected.
RESPONSE: Thank you for your suggestion. We have ensured that all sequences used in this comparative analysis are properly referenced. All sequences that appeared in the NCBI GenBank as of the accession date (January 9th, 2025) were included in the analysis. However, incorporating them into Table 2 leads to oversaturation, making interpretation difficult. Therefore, we have decided to maintain the table in its current form.
Point 4: Information about bioethics (Line 64-68) should be provided at the end of the article in the appropriate section. It is unnecessary here.
RESPONSE: We sincerely appreciate your thoughtful suggestion. However, we believe that including information about the bioethics approval concerning the sample collection is a related matter.
Point 5: Line 12 – correct word – “hosts”. It is necessary to add information about the intermediate hosts of these parasites here.
RESPONSE: We acknowledge and appreciate your remark. However, it would be too difficult to list all of the intermediate hosts of Sarcocystis spp. in the abstract due to journal requirements, as abstract could contain not more than 200 words
Point 6: The first table in the results is not needed. Part of the numerical material is already given in the text of the article. The remaining figures should also be given in the text. Then the numbering of the tables in the article will be corrected ?
RESPONSE: Your suggestion was greatly appreciated as we deleted Table 1 from the results part and indicated all the necessary information in the text.
Point 6: At the first mention of genus or species, its full Latin name with the author and year of description should be given; in relation all animal species and Sarcocystis spp. (lines 26, 35, 55, 56, etc.). This should also be done in Tables 1 and 3. In Abstract Latin species name must be written in full.
RESPONSE: Thank you for your suggestion. We have made the appropriate revisions accordingly.
Point 7: Authors must submit the manuscript in accordance with the rules for formatting MDPI articles (Article title, all words in which must be capitalized).
RESPONSE: Thank you for your valuable input. The title of the article was changed accordingly.
Round 2
Reviewer 1 Report
Comments and Suggestions for Authors
- For the Introduction, the authors did not mention the reason why raccoon dogs are suspectable to pathogens. Please add a paragraph regarding how raccoon dogs play as an intermediate or reservoir for pathogens, including viruses, bacteria and parasites using the following reference papers (https://doi.org/10.3390/pathogens13030270; https://doi.org/10.1186/s13071-022-05245-3; https://doi.org/10.1371/journal.ppat.1012204; https://doi.org/10.1016/j.ijppaw.2021.09.008).
- For the Discussion, the authors did not bring up the life style of Sarcocystis regarding raccoon dogs and human (if possible). Please add a paragraph and figure (strongly suggested) describing how Sarcocystis is crucial to the natural intermediate (such as cervids, wild boars, pigs, goats and sheep) that you mentioned in the abstract.
Author Response
Point 1: For the Introduction, the authors did not mention the reason why raccoon dogs are suspectable to pathogens. Please add a paragraph regarding how raccoon dogs play as an intermediate or reservoir for pathogens, including viruses, bacteria and parasites using the following reference papers (https://doi.org/10.3390/pathogens13030270; https://doi.org/10.1186/s13071-022-05245-3; https://doi.org/10.1371/journal.ppat.1012204; https://doi.org/10.1016/j.ijppaw.2021.09.008).
RESPONSE 1: Thank you for your valuable remark. We have added information about the raccoon dogs’ susceptibility to pathogens and its role as a spreader of many different bacteria, viruses, parasites, etc.
Point 2: For the Discussion, the authors did not bring up the life style of Sarcocystis regarding raccoon dogs and human (if possible). Please add a paragraph and figure (strongly suggested) describing how Sarcocystis is crucial to the natural intermediate (such as cervids, wild boars, pigs, goats and sheep) that you mentioned in the abstract.
RESPONSE 2: Thank you. We have added information about zoonotic Sarcocystis spp. in humans and negative impact of acute infections of various Sarcocystis spp. in farmed animals and cervids. Newly created figure is also present in the article.
Round 3
Reviewer 1 Report
Comments and Suggestions for Authors
The authors addressed all my questions. Thanks for your time.